# *Chile* (*Capsicum* spp.) as Food-Medicine Continuum in Multiethnic Mexico

**DOI:** 10.3390/foods10102502

**Published:** 2021-10-19

**Authors:** Araceli Aguilar-Meléndez, Marco Antonio Vásquez-Dávila, Gladys Isabel Manzanero-Medina, Esther Katz

**Affiliations:** 1Centro de Investigaciones Tropicales, Universidad Veracruzana, Xalapa 91050, Mexico; 2Tecnológico Nacional de México, Campus Valle de Oaxaca, Oaxaca 71230, Mexico; marco.vd@voaxaca.tecnm.mx; 3Centro Interdisciplinario de Investigación para el Desarrollo Integral Regional Instituto Politécnico Nacional, Oaxaca 71230, Mexico; 4Institut de Recherche Pour le Développement (IRD), UMR 208 PALOC IRD/MNHN, MNHN, 75005 Paris, France; esther.katz@ird.fr

**Keywords:** *Capsicum*, food-medicine continuum, indigenous communities, Afrodescendants, body healer

## Abstract

Mexico is the center of origin and diversification of domesticated *chile* (*Capsicum annuum* L.). *Chile* is conceived and employed as both food and medicine in Mexico. In this context, the objective of this paper is to describe and analyze the cultural role of *chile* as food and as medicine for the body and soul in different cultures of Mexico. To write it, we relied on our own fieldwork and literature review. Our findings include a) the first matrix of uses of *chile* across 67 indigenous and Afrodescendants cultures within Mexican territory and b) the proposal of a new model of diversified uses of *chile*. Traditional knowledge, uses and management of *chile* as food and medicine form a continuum (i.e., are not separated into distinct categories). The intermingled uses of *Capsicum* are diversified, deeply rooted and far-reaching into the past. Most of the knowledge, uses and practices are shared throughout Mexico. On the other hand, there is knowledge and practices that only occur in local or regional cultural contexts. In order to fulfill food, medicinal or spiritual functions, native communities use wild/cultivated *chile*.

## 1. Introduction

*Chile* (common Spanish name derived from the Nahuatl language to be used here) is native to the Americas; however, it now flavors cuisines around the globe [1,2,3]. Mexico is considered the center of origin and diversity of domesticated *Capsicum annuum* L. *chile* with a continuous use of wild and domesticated varieties of at least 6000 years [4,5,6]. Thus, *chile* along with maize (*Zea mays*), beans (*Phaseolus* spp.) and squash (*Cucurbita* spp.) have co-evolved with local groups even before the establishment of civilizations was recognized [7]. Currently, in Mexico there are about 60–70 indigenous groups that have safeguarded traditional/local environmental knowledge (TEK) related to the management, production and diversified use of natural resources including *chile* [8,9]. Two works framed in Mexico have described various uses of *chile* but were not exhaustive. Long-Solís [10] described various cultural aspects of *chile* at different times in the winding and intricate history of this region of the world. Data from Aguilar’s doctoral dissertation [9] also describe other linguistic, historical, and archaeological data on the versatility and complexity of cultural patterns of *chile* production and use.

Due to the economic and food importance of *chile* around the world, hundreds of studies have been conducted on the nutraceutical importance of the spicy fruits when consumed directly in fresh bites or as part of a dish prepared with fresh or dried *chile* (without industrial additives) in daily or festive food preparations throughout the world [11,12,13]. The use of *chile* as a food emerged in the Americas and with the exchange of plants and global cultural processes that occurred 500 years ago, this plant spread to virtually all continents [1,14]. Today, in many Asian, African, European, and American countries, *chile* is part of the daily diet and is even an essential ingredient in emblematic dishes such as the masalas in India, goulash in Turkey, curries in Thailand, etc. [15,16]. Nowadays, there is a whole food industry that uses technology to transform *Capsicum* fruits into flavoring, coloring and preservation of other foods that are produced on a large scale and distributed over long distances [17,18,19]. Therefore, *chile* are in great demand depending on the purpose for which they are used worldwide.

*Chiles* are utilized for many purposes. They pertain to many varieties, mainly of *C. annuum* [20]. However, few studies of this species have adopted an ethnobotanical approach in order to analyze ethnographic aspects in the center of origin and diversity. These indicate ancient/modern uses of *chile* in each of the indigenous cultures within a multiethnic territory where the fruit is a deeply rooted element of intensive daily use. For example, there are studies of the Nahua of the Huastec region, Tlapanecs of Guerrero, Mixtecs and Zapotecs of Oaxaca where *chile* is the protagonist, most of them documented by the authors of the present study [21,22,23]. There are other biocultural studies of the past that only mentioned *chile* as part of a list [24].

Currently, with a disciplinary vision within the hard sciences, modern uses of *chile* for medicinal purposes have also been validated in a scheme unrelated to traditional Mexican medicine [25]. These studies explored the effectiveness of functional properties often attributed to carotenoids, vitamin C and E, alkaloids, flavonoids, and capsaicin that support the maintenance of health and well-being [11]. However, by eliminating *chile* and analyzing them separately from their primary cultural context, the complex biocultural relationships that are part of the cultural construction of health-related ideas and practices are occult for the academic eye.

It stands out that the secondary metabolites of the spicy fruits have been managed, selected, and diversified for use as food and/or medicine without interruption by different cultures of Mexico [26]. Indigenous territories of Mexico have been culturally divided into Mesoamerica (central and southern Mexico) and Arid America (northern Mexico). In Mesoamerica there is a particular conceptualization of the natural world as a living being that holds intimate relationships of exchange and mutual dependence with the social, human world [10,11]. This interdependence is expressed in the multifaceted way in which people interact with and use biological resources. Within the same region, health is understood as being the combination of physical, mental, social, and spiritual well-being while life is thought of as a combination of body, mind, and soul. The Nahuatl terms refer to major concentrations of soul forces and vital fluids distributed throughout the organism, but concentrated in the head, heart, and liver, respectively [12,13]. The soul or spirit of a person can be affected by various natural and supernatural agents, causing psychosocial and somatic imbalance. The ailing person then turns to traditional medicine to regain balance, health, and well-being. In practically all ancient cultures as well as their modern descendants, healthcare was a concern that remained mainly on the family level, based on basic knowledge linked to the use of medicinal herbs, roots, and minerals, which could be acquired without any problem at markets [27,28]. In addition, there are specialists that utilize herbalism, bone manipulation, various basic surgical techniques, divination and incantations in their practice [28,29]. Under this cosmology, traditional medicine can be preventive or corrective. Therefore, it seems relevant to highlight that *chile* has been used as a food–medicine continuum as mentioned by other resources elsewhere [30,31,32,33]. In this context, the objective of this paper is to describe and analyze the cultural role of *chile* as food and as medicine for the body and soul in the different cultures of Mexico.

## 2. Materials and Methods

### 2.1. Study Area

Mexico is the study area and it has a population with blurred ethnic boundaries [34] (see Figure 1), as other Latin American countries. In 2015, speakers of indigenous languages represent 6.5% of the total population in Mexican territory [35], but many people considered today as descend from indigenous people who lost their language [36,37]. They often share many cultural elements with the native communities of the same region, including agricultural and culinary traditions. The names of the native languages spoken were taken as a reference to delimit the modern cultures which descended from Mesoamerican intellectual traditions [38]. The list is based on the catalogue of indigenous languages of Mexico [39] that counts 67 linguistic groupings, used here as a unit of study, distributed in 364 linguistic variants (formerly linguistic groupings were considered as languages and their variants as dialects, but linguists have demonstrated that these last ones were actually languages). In addition, for the current analysis of the human-*Capsicum* relationship in Mexico, two groups of Afro-descendants were considered (one of Guerrero and Veracruz, and the Mascogo community of Coahuila).

### 2.2. Field Study

Araceli Aguilar has collected ethnobotanical data over 20 years, focusing on indigenous territories, following The Code of Ethics of the International Society of Ethnobiology. Datapoints of botanical samples gathered from fieldwork, national herbarium specimens and other datasets such as the CONABIO were utilized to draw the map of Figure 1. The map was already published (with graphical variation) in previous studies [20]. The others authors recovered ethnobotanical and ethnographic information regarding the genus *Capsicum* while focusing on the ethnoecology of *chile*, nurse plants, fowl, and Mexican indigenous people, such as the Chontales of Tabasco, the Zapotecs of Guelavía, and the Mixe of Guichicovi, Oaxaca (MAVD); working on Zapotec home gardens, ethnobotany of traditional agroforestry systems and traditional markets in Oaxaca (GIMM), and on food and agriculture with Mixtec people in Oaxaca and food studies around the country (EK).

### 2.3. Bibliographic Research

The authors reviewed articles of natural sciences, articles, books, and thesis of social sciences and cookbooks of the CONACULTA (cultural branch of Mexican federal government) collection called *Cocina indígena y popular* (Indigenous and popular cuisine). The search for natural sciences articles (original or review) was performed using the PubMed, Google Scholar, and Scopus electronic databases until March 2021. The main keywords used were: “*Capsicum* medicine”; “traditional use of *Capsicum*”; “Mesoamerica and *Capsicum*”; among others. All the included references were manually selected, reviewed, and added to the database in Filemaker by the authors.

### 2.4. Analysis of Cookbooks

In total, 77 cookbooks from the series *Cocina indígena y popular* and other Spanish written documents were reviewed. A database was constructed to determine the possible variety by inferring it with the common name. Recipes and culinary processes were registered as well. Annotations about the languages were important to determine the cultural context of the cookbook.

### 2.5. Classification of Illnesses

Diseases or symptoms associated with potential modern diseases were classified according to the 2021 WHO system, the ICD-11 version 09-2020 (https://icd.who.int/browse11/l-m/en (accessed on 1 March 2021)), into 12 categories: (1) infectious or parasitic diseases, (2) mental, behavioural, or neurodevelopmental disorders, (3) diseases of the nervous system; (4) diseases of the visual system; (5) diseases of the ear or mastoid process; (6) diseases of the respiratory system; (7) diseases of the digestive system, (8) diseases of the skin; (9) diseases of the musculoskeletal system or connective tissue, (10) diseases of the genitourinary system, (11) pregnancy, childbirth or puerperium and (12) poisoning. This classification is an interpretation/translation of the authors of this paper and is not based on an emic view of the local medicine of each culture. Medicine for the soul area: to describe cultural-bound syndromes that Western mental health professionals and scientists fail to validate as “real illnesses” [40].

## 3. Results

*Chile* functions as a multifocal symbol operating intimately on many levels of people’s lives. Plant use is an integral part of the mental and physical life of people who live in direct contact with their natural resources. This is a resilient factor among indigenous communities worldwide. Like another Latin American countries, Mexico has a mixed population with blurred ethnic boundaries [34]. Presently speakers of indigenous languages represent 6% of the population [41], but many people considered today as mestizos descend from indigenous people who lost their language [36,37]; they often share many cultural elements, including agricultural and culinary traditions, with the native communities of the same region. *Chile* are always part of all of them since it is a crop that has intertwined its history with the history of the cultures that domesticated it [38].

The analysis of 77 cookbooks edited by CONACULTA gave some clues as to which *chile* are depicted on modern written documents (Table 1). Only six cookbooks were bilingual; Nahuatl [42], Tarahumara [43], Mayo [44], Yaqui [44], Purepecha [45], Totonac [46], Tepehuan [47] and also included common names for *chile* on each languages. In total, 32 out of 77 have some common names for ingredients in local languages but not terms referring to *chile* were included. Culinary processes mentioned for cooking or using *chile* were: deveined, roasted, boiled, grilled, ground, *molcajeteado* (crushed in a stone such as mortar), chopped, sieved, crushed, seeded, cut into small pieces, washed, blended, stuffed, stuffed, fried, cooked, shredded, scrambled, burned, soaked, ground, parboiled, strained, soaked, browned in oil, ground in *metate*, sprinkled, filleted, finely chopped, stewed, seasoned, sautéed, mashed, chopped, sliced.

**Table 1 foods-10-02502-t001:** Use of *chile* as food, medicine for the body and the soul in Mexico.

Family language	Language/Cultural Group	Food	Medicine for the Body	Medicine for the Soul
Oto-manguean	Otomi	[48]		[10]
	Mazahua	[49]		
	Matlatzinca	[49]		
	Tlahuica	[49]		
	Pame	[50]		
	Chichimeco jonaz	[51]		
	Chinantec	[52]		[10]
	Tlapanec	[53]		[54]
	Mazatec	[55]		
	Ixcatec	[56]		
	Chocholtec	[57]		[57]
	Popoloca	[58]		
	Zapotec	[23,59,60]	[61]	[23]
	Chatino	[62]		
	Amuzgo	[53]		
	Mixtec	[22,53,63]		[22]
	Cuicatec	[64]		
	Triqui	[65]		
Maya	Tenek	[29]	[29]	[29]
	Maya Yucatec	[66,67,68]	[69]	[69]
	Lacandon	[70]		
	Ch’ol	[71]		
	Chontal of Tabasco	[72]		
	Tseltal	[73]		
	Tsotsil	[73]	[74]	[74]
	Q’anjob’al	[75]		
	Akateko	[76]		
	Jakalteko	[77]		
	Qato’k	[78]		
	Chuj	[79]		
	Tojolabal	[73]		
	Q’eqchi’	[80]		
	K’iche’	[81]		
	Kaqchikel	[81]		
	Teko	[75]		
	Mam	[73]		
	Awakateko	[82]		
	Ixil	[83]		
Totonaco-Tepehua	Totonac	[46]	[46]	[46]
	Tepehua	[84]		
Purepecha	Purepecha	[45]		[45]
Mixe-zoque	Mixe	[85]	[86]	
	Sayulteco	[87]		
	Oluteco	[88]		
	Texistepequeño	[89]		
	Ayapaneco	[90]		
	Popoluca	[60]		
	Zoque	[70]		
Chontal de Oaxaca	Chontal of Oaxaca	[91]		
Huave	Huave	[92]		[93]
Algica	Kickapoo	[94]		
Yuto-nahua del sur	Pápago	[95]		
	Pima	[96]	[96]	
	Northern Tepehuan	[47]		
	Southern Tepehuan	[47]		
	Tarahumara	[43]	[97]	
	Guarijio	[98]		
	Yaqui	[44,99]		
	Mayo	[44]		
	Cora	[100]		
	Huichol	[101]		
	Nahuatl	[53,60,102]		[21]
Yuto-nahua del sur	Paipai	[103]		
	Cucapá	[103]		
	Kumiai	[103]		
	Kiliwa	[103]		
Seri	Seri	[104]		
No language affiliation	Mascogos	[105]		
No language affiliation	Afrodescendants	[106,107]		

Out of 3641 records, 561 *chile* with no specific name, 105 wild *chile,* and 2975 domesticated *chile* were used. Three main groups of *chile* were commonly used in recipes all over Mexico. The grouping was made by the fresh and dry state of the same variety (except for *guajillo*) (1) complex *Poblano*/*Ancho*/*Color* (550 records); (2) *Guajillo* (445 records) and (3) *Jalapeño*/*Chipotle* (337 records), common names for fresh *chile* of this group are: *Jalapeño, Cuaresmeño, Gordo* and *Huauchinango*, common names for dry *chile* are: *Chipotle, Mora* and *Morita*. Another classification was the rarely mentioned names (44 records with less than 3 records), 89 *chile* are *C. chinense* and *C. pubescens* versus 2535 corresponding mainly to domesticated varieties of *C. annuum*. The variability of the amount of spice consumed in Mexico’s culinary mosaic is impossible to describe by scientific standards. Some studies have made metabolomics studies that give some hints as to the biochemical variants expressed in the flavor, heat, and odor of *chile* [108] but tests have yet to be conducted on the complex culinary dishes of multi-ethnic Mexico.

### 3.1. Chile as Food

*Chile* is ubiquitous in all Mexican food. According to Elisabeth Rozin [109], it is its “flavoring principle”. However, it is much more than a flavor. For Mexican people, most foods cannot be eaten without *chile*. Not only would it be bland, but also it would not be a balanced meal. Most Mexicans conceive that hot and cold principles rule food and body health. According to cultures and regions, different qualities may be attributed to foodstuffs, but they are usually the same for the staples: corn is considered as warm, beans as cold and *chile* is hot. “Cold” food is not good for the body, a meal must tend toward “warm” food, without being too hot, except in case of a “cold” disease, when food must heat up the body [110].

For centuries, Mexican peasants have been basing their diet on boiled beans and corn tortillas, with *chile*. For city dwellers, corn and beans may not be the basis of their diet, if they consume more meat, for instance, but are at least the side dish. Corn and beans balance each other, beans cannot be eaten alone, and usually are cooked with *chile*, as “cold” food is supposed to be heavy to digest. Then raw *chile* or a *chile* sauce is served on the table, and each eater adds it to the plate according to the individual taste. Most dishes are cooked together with raw or dry *chile* (meat, vegetables, etc) or in elaborated *chile* sauces (*mole, adobo*, etc). *Chile* is added to industrial food, such as tortilla chips, to sweets, even when designed for children. Corn on the cob and sliced fruits (such as mango) eaten as snacks, and often sold on the street, are served with lemon and *chile* powder. Without *chile*, the body balance would be in danger. However, if the person is sick, *chile* is not recommended [102]. Many people we interviewed declared that they were strong or resistant thanks to *chile*. This was particularly true in rural areas, where people compared themselves to city dwellers, or to “gringos”, whom they judged weaker. Children get used to *chile* from their earliest age, in their mother’s womb, through their mother’s milk, from the smells of their family’s kitchen. *Chile* is progressively introduced in their food so that at an early age (some beginning at the age of four), children like to eat *chile* [111].

### 3.2. Chile as Medicine for the Body

Due to the effectiveness of using *chile* to heal, dozens of modern scientific studies have currently been carried out to recognize its therapeutic action [112,113,114,115] and that have indirectly validated the ancestral and traditional uses. Here we depicted a brief description:(1)Certain infectious or parasitic diseases: antibacterial and antimicrobial [96,116]; the Pimas of Sonora use it as antibacterial [96]; Nematicide [10,117]: the Pimas of Sonora use it as a nematicide [96]; Fever: in Santo Domingo Petapa and Santa María Petapa, the Zapotecs of the Isthmus to relieve fever [61]; the korí chókame (black chile) of the Raramuris is used to prepare a tea to relieve fever [97]; in Santo Domingo Petapa and Santa María Petapa, the Zapotecs of the Isthmus use chile to relieve fever [61].(2)Mental, behavioral, or neurodevelopmental disorders: the Raramuris and Mestizos use isíburi (plant mixture with chile) to treat hangovers and fever [97].(3)Diseases of the nervous system. The BDMTM mentions that it is used as an antineuralgic [118]; the Raramuris use a mixture of chiltepín (C. annuum var. glabriusculum) with chupate (Ligusticum porter, Apiaceae) to treat headaches; a BDMTM mentions that it is used as an antineuralgic [118].(4)Diseases of the visual system [61,112,114]: the Mestizos of Querétaro State use it to relieve an eye infection by applying a crushed quipín chile (C. annuum var. glabriusculum) [119]; in Santo Domingo Petapa and Santa María Petapa, the Zapotecs of the Isthmus apply chile leaves for eye problems [61]; Tenek of San Luis Potosí use it to cure eye problems [29].(5)Diseases of the ear or mastoid process [61,112,114]: Sahagún quoted by López Austin [120] says that the Nahua cured ear ulcers with warm drops of coyoxóchitl with chile; the Raramuris use it to treat ear pain [97]; mestizos from Sonora use the oil of chiltepín (C. annuum var. glabriusculum) to cure ear pain [121].(6)Diseases of the respiratory system [46,69,113,120]: Nahua to treat cough in Tlanchinol, Hidalgo [122]; mestizos in the State of Querétaro cure certain pulmonary ailments and fevers by smoking dried chile to cause sweating and coughing [119]; Sahagún quoted by López Austin [120] says that the Nahua drank water from the root of the tlacopópotl, and lime water with chile and a decoction of iztáuhyatl to cure coughs and expel phlegm. Mestizos from Sonora eat a lot of chiltepín (C. annuum var. glabriusculum) to avoid the flu and smoke it with tobacco to remove the cough [121].(7)Diseases of the digestive system including teeth conditions [69,113,120,123]; mestizos in the State of Querétaro use it to prevent constipation and gastritis [119]; Sahagún quoted by López Austin [120] says that the Nahua cured constipation by administering through the anus a suppository made with soot and a little saltpeter, kneaded with rubber filled with chile, made into a ball and inserted from behind, and to tartar diarrhea they drank chía (Salvia hispanica) atole (maize based drink) mixed with chía totopos (grilled tortillas) and sprinkled with chile; Mestizos from Sonora use chiltepín to cure ulcers, gastritis, and hemorrhoids [121]; the BDMTM mentions that it is used as an antidiarrheal, carminative, eupeptic [118]; Lacandon Maya uses chile to soothe toothache and inflamed gums [124].(8)Diseases of the skin [113]: mestizos in the State of Querétaro apply an patch of ground piquín chile to areas of the skin affected with erysipelas or festering so that the wound does not become infected and that it helps with pain [119]; mestizos in Sonora use chiltepín (C. annuum var. glabriusculum) to heal wounds [121]; the BDMTM mentions that in Veracruz and Oaxaca it is used to treat chincual de criatura (cultural illness), erysipelas and wounds, as well as being used as an antiseptic [118]; Tenek of San Luis Potosí use it to cure skin problems [29]; the Zapotecs of San Juan Guelavía use it roasted and dried to remove pimples from the face [23]; in the Tzotzil of Zinacantán, Chiapas [74].(9)Diseases of the musculoskeletal system or connective tissue: the BDMTM mentions that it is used as antirheumatic [118]: the Raramuris use chiltepín in poultice to relieve arthritic pain in the hands [97]; to cure broken bones: Otomí bonesetters use to massage the rib cage while the patient blows into a bottle to reset broken ribs. Then they put a cataplasm of sacasil tuber ground with cumin (Cuminum cyminum), cayenne chile (Capsicum sp.), and cloves (Eugenia caryophyllus) over the lesion [125].(10)Diseases of the genito-urinary system; [69,113,120].(11)Pregnancy, childbirth, or puerperium help to mitigate labor pain and promotes delivery [69].(12)Poisoning. Mestizos from Sonora use chiltepín (C. annuum var. glabriusculum) with tallow for tarantula bites [121].

### 3.3. Chile as Medicine for the Soul

The use of herbal medicines, and complementary and alternative medicine, is increasing globally. There is an emphasis on a holistic approach, consistent with definition of health of the World Health Organization (WHO). Health is understood as being the combination of physical, mental, social, and spiritual well-being and life as being a union of body, mind, and soul. In this section, we will describe *chile* as medicine for the soul. The concept of the soul is not universal, nor is it found in all cultures. In Mexico, several native terms are translated as soul, such as *tonalli* among the Nahua [126] or *pixan* among the Maya [127]. According to the Nahua, the fetus receives both a soul and a heart during the gestation process. The soul is given by the Sun-God and is therefore considered “warm”; based in the heart, it endows the infant with life and the capacity for movement and growth [128]. The soul or spirit of a person can be affected by various natural and supernatural agents, causing psychosocial and somatic imbalance. The sick person then turns to traditional medicine to regain balance, health, and well-being. Many traditional cultures in Mexico use diverse species of plants (and a few animals) to cure people of the evil eye (*mal de ojo* in Spanish). Generally, it can be characterized as the personal emanation of a force that arises involuntarily due to a strong desire and that harms the desired being. The evil eye is a recognized cause of health care in Mexico, especially by rural inhabitants and traditional healers. This condition is considered a culture-bound syndrome in which there is a state of restlessness characterized by anxiety, depression, insomnia, and loss of appetite [129]. Children are the main population affected by the evil eye. *Chile* is used in various ways to prevent and cure the evil eye in several ethnic cultures of Mexico; for example, the Zoques of Chiapas [130], the Zapotecs of Oaxaca [23], five ethnic groups of Puebla [128], the Huastecs of Veracruz [131], the peasants of Querétaro [119], and the Guarijío and Mayo of Sonora [121]. Other ailments related to the soul and in which *chile* is used as medicine are: “dead people’s evil wind” (*mal aire de muerto* in Spanish), a bad entity that comes from people who just died, registered in Oaxaca with Mixes and Mixtecs [22,132]; evil supernaturals, fright, and witchcraft among Zapotecs of Oaxaca [23,133].

### 3.4. A Model of Human Management of Chile in Mexico

The model of human management of *chile* proposed here is shown as a continuum that goes from wild to domesticate with blur limits (Figure 2). As an addition to this model, the native species utilized are wild and domesticates of *C. annuum* and *C. frutescens*.

There is a wide range of uses where the spicy fruit is consumed and putatively act as a protector element, medicine, and/or food. A more holistic way of presenting natural resources within the academic world is increasing in popularity [134] recognizing the need to present a more “real” picture of how natural resources are perceived and utilized.

## 4. Conclusions

This work shows that Mexico’s ethnic cultures conceive, use and manage *chile* as a food-medicine continuum and can possibly be explained because it is a fruit that, similar to corn, has a deep symbolic meaning in Mesoamerican and Arid American cultures.

The use of *chile* in food is common to the whole territory providing nutrients to the Mexicans who consume them [25]. The present study shows data confirming this fact. However, regional studies that connect the studies of secondary metabolites of local cultivars with the preferences of the uses of each culture are needed to achieve a better understanding of the patterns of human selection on a plastic crop of the Solanaceae family. This study also concluded that medicinal uses for the body and soul have very restricted use in current times when compared to the presence of *chile* as food.

We need to further test the local concepts of illness (imbalance) and health (equilibrium) using *chile* as a theoretical axis to approach descendants of the Mesoamerican intellectual traditions within the multiethnic territory. *Chile* has been used in different cultural spaces at different times to cure various diseases that were classified here in 12 categories. Scientific results validated many of the ancient uses of *chile* as medicine (see Table 1 and section *chile* as medicine for the body). Some questions to be pursued for future studies are (1) have medicinal uses been lost at different scales (community, language/culture, or nationwide)? (2) do we have knowledge of the ingredients, preparation processes, and utensils to prepare *chile* as a medicine for a particular ailment?

Although there are many studies that validate the efficacy of *chile* in curing diseases such as cancer [135]. Other healing properties have been attributed to *chile*, as documented in this study, particularly the properties to cure culturally affiliated diseases whose biochemical foundation is unclear or it might be healing in other forms. Eleven ethnic groups of Chiapas, Oaxaca, Puebla, Veracruz, Querétaro, and Sonora use *chile* to prevent and cure the evil eye. In Oaxaca, Mixes, Mixtecs, and Zapotecs use *chile* against evil beings, fright, and witchcraft. Therefore, we should initiate systematic studies in Mesoamerican cultural contexts to acknowledge more complex illnesses that have been made invisible for the global academic world.

There are shared uses, practices, and knowledge throughout the Mexican territory such as the spicy fruits being used as condiments and vegetables to prepare food. On the other hand, specific uses within the medicinal (body and soul) categories are found in local contexts unique to a particular indigenous culture that are not repeated in the rest of the country. It may be the reason why there is a high diversity of *chile* in the form of cultivars of *C. annuum* var. *annuum* that are the result of locally selected unique morphotypes.

Further research is needed to fill the gaps in local and indigenous knowledge by training local ethno-scientists [136] to carry out participatory research with a dual purpose: to generate academic knowledge and to restitute that knowledge to the communities where biodiversity is being conserved [137]. This is particularly relevant in one of the most megadiverse countries of the world, Mexico.

## Figures and Tables

**Figure 1 foods-10-02502-f001:**
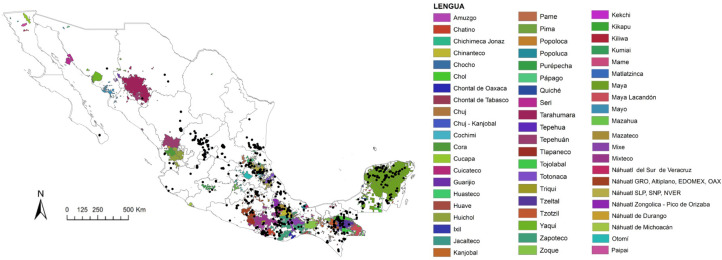
Distribution map of *Capsicum annuum* var. *annuum* in Mexico in relation to the indigenous cultures. The colored backgrounds indicate the territories of the 67 indigenous languages spoken in Mexico. (Map elaborated by A. Aguilar-Meléndez and Andrés Lira-Noriega from personal and CONABIO databases, and for the Amerindian territories, from maps provided by INALI and the anthropologist Eckart Boege).

**Figure 2 foods-10-02502-f002:**
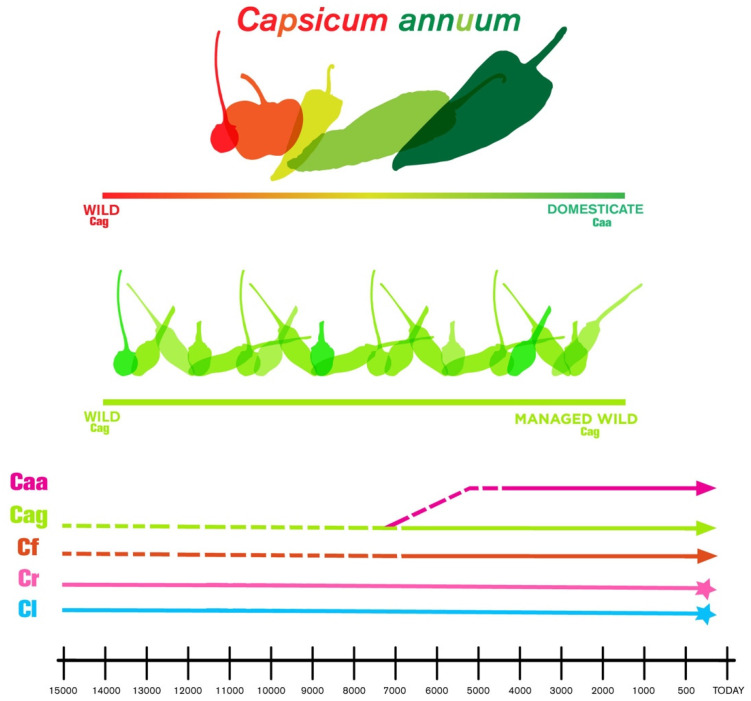
Model of human management of *chile* in Mexico. *C. lanceolatum* (Cl) and *C. rhomboideum* (Cr) were never utilized by humans (blue and pink ending in a star); *C. frutescens* (Cf) that includes wild and domesticates; *C. annuum* var. *glabriusculum* (Cag) are the wild *chile* and putative ancestor of Caa, and *C. annuum* var. *annuum* (Caa) includes the modern domesticated landraces and commercial varieties.

## Data Availability

Not applicable.

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
