# Peer review of "Chile (Capsicum spp.) as Food-Medicine Continuum in Multiethnic Mexico"

_foods, 2021, doi:10.3390/foods10102502_

Round 1
Reviewer 1 Report
I think this manuscript is a review manuscript and not a research article.
In the abstract authors should better limit the area considered for the study (name of the cultivation area should be added).
Too poor introduction about the state of art (add more references); authors should focus on the main research topics and relevant questions to be addressed. Some reference should be added in relation to phytochemicals in Capsicum spp. (previous studies?) and relative properties, and the influence of agro-environmental conditions on their concentration.
Materials&Methods section should be reduced in order to give to the readers only the necessary information to understand the study and the results.
Results should be integrated in the Discussion section to avoid repetitions in the text.
Conclusions should be added to the text. It may be useful to summarise the main results in relation to the aims of the manuscript.
Author Response
10 01 2021 Review of Chile for Foods
Open Review 1
- Comment of reviewer 1: I think this manuscript is a review manuscript and not a research article.
- Answer: This paper has mix data from fieldwork and bibliographic research. It is a research article based on ethnobotanical approach.
- Comment of reviewer 1: In the abstract authors should better limit the area considered for the study (name of the cultivation area should be added).
- Answer: The summary was rewritten at the suggestion of the reviewer. In addition, a graphical summary was prepared
- Comment of reviewer 1: Too poor introduction about the state of art (add more references); authors should focus on the main research topics and relevant questions to be addressed. Some reference should be added in relation to phytochemicals in Capsicum (previous studies?) and relative properties, and the influence of agro-environmental conditions on their concentration.
- Answer: The introduction was completely rewritten, and more references added to highlight ethnobotanical points relevant to the study area. See new version of the paper.
- Comment of reviewer 1: Materials & Methods section should be reduced in order to give to the readers only the necessary information to understand the study and the results.
- Answer: The materials & methods was rewritten using subheadings to organize the different methods and to describe every section in more detailed manner. A map was added to delimit our area of study and to show the languages/cultures distribution.
- Comment of reviewer 1: Results should be integrated in the Discussion section to avoid repetitions in the text.
- Answer: Results and discussion sections were merged. See new version of the paper
- Comment of reviewer 1: Conclusions should be added to the text. It may be useful to summaries the main results in relation to the aims of the manuscript.
- Answer: Conclusions were added. See new version.

Reviewer 2 Report
Very interesting manuscript. It requires a few corrections, which are more of a cosmetic.
Below are my comments to the authors:
1. I would change the title slightly because it is quite confusing in relation to the content.
2. Modify your keywords, the article is perhaps more interesting.
3. Abstract - especially its second part needs to be improved - poorly reflects what is at work.
4. The materials and methods section is hard to repeat - please add details slightly.
5. The discussion needs to be improved - it is too short so far.
6. The summary must be rewritten - after the work you can see that the authors can afford much more.
7. Literature can be supplemented - there are many more works to be cited on the Internet.
Generally, the work is worth publishing and the corrections are rather cosmetic.
Author Response
Open Review 2
- I would change the title slightly because it is quite confusing in relation to the content. Answer: The title was changed. Thank you for the suggestion.
- Modify your keywords, the article is perhaps more interesting. Answer: they were changed. See the new version of the paper.
- Abstract - especially it is second part needs to be improved - poorly reflects what is at work. Answer: it was changed. See the new version of the paper.
- The materials and methods section is hard to repeat - please add details slightly. Answer: The materials & methods were rewritten using subheadings to organize the different methods and to describe every section in a more detailed manner. A map was added to delimit our area of study and to show the languages/cultures distribution.
- The discussion needs to be improved - it is too short so far. Answer: Results and discussion sections were merged. See the new version of the paper.
- The summary must be rewritten - after the work you can see that the authors can afford much more. Answer: The summary was rewritten at the suggestion of the reviewer. In addition, a graphical summary was prepared
- Literature can be supplemented - there are many more works to be cited on the Internet. Answer: More references were added to highlight ethnobotanical points relevant to the study area. See the new version of the paper.

Reviewer 3 Report
Foods’s manuscript
Titled:
Chile (Capsicum spp.) as spice, protector and soul/body healer.
General comments:
-In general, this manuscript has a valuable topic. The topic is scientifically sound.
- The manuscript is well written except for minor English language check required.
-The experimental design is adequately discussed.
- My main concerns were the abstract, introduction and the discussion section.
-There are some minor comments.
Detailed comments:
-In general, please avoid using personal pronouns such as we, our, and apply this rule throughout the manuscript for example We in (Line20 and line 322).
Abstract:
The authors didn’t provide the direct aim of the study.
This section is not properly written. It needs some modifications and needs to be more descriptive and informative about the current study.
Please include some important values in the abstract from the results that presents the most significant findings
Keywords:
Please add Body healer to the keywords list.
Introduction:
This section doesn’t provide enough background about the topic. I see that the introduction is very poor. This section needs to be enriched and provided with more background about the topic.
Materials and Methods:
The experimental design and the materials and methods section were Not adequately described.
Please rewrite this section, use subheadings to organize the different methods and describe all methods in more detailed manner.
Results:
-The authors did a good job in this section. The results were well presented But I think the author mixed up the discussion section here in the results although he made a separate section for the discussion that was poorly written.
Discussion:
I had a hard time to make the connection between your discussions and the results in which figure. The author should mention data in figure and tables during the discussion.
Please make subheadings in this section in organized way, the same order as the subheadings in the results section.
-Please rewrite this section in more organized way and relate to the data in tables and figures carefully with a comparison to the previous studies.
*If it is easier to discuss the results clearly and thoroughly, the author is advised to combine the results and the discussion in one section for the best presentation and discussion to the results.
Conclusion:
This section is missing. Although this section is nod mandatory, it is better to add a conclusion section to this manuscript. Please provide a good conclusion for the study and include the significant findings. This section must be supported by the results.
References:
Although, the author provided enough citations, the author is advised to include more recent citations (The last 5 years work).
*I am convinced that this manuscript is very valuable.
Author Response
Open Review 3
Reviewer 3: In general, please avoid using personal pronouns such as we, our, and apply this rule throughout the manuscript for example We in (Line20 and line 322). Answer: Comparing with Teixidor-Toneu et al. 2021 that was the first paper published within the series of “Plant food-medicines: perceptions, traditional uses and health benefits of food botanicals, mushrooms, and herbal teas”, we prefer to write using the grammatical form of first-person.
Reviewer 3: Abstract: The authors didn’t provide the direct aim of the study. This section is not properly written. It needs some modifications and needs to be more descriptive and informative about the current study. Please include some important values in the abstract from the results that present the most significant findings. Answer: The summary was rewritten at the suggestion of the reviewer. In addition, a graphical summary was prepared and added. See new version.
Reviewer 3: Keywords: Please add Body healer to the keywords list. A body healer was added.
Reviewer 3: Introduction: This section doesn’t provide enough background about the topic. I see that the introduction is very poor. This section needs to be enriched and provided with more background about the topic. Answer: The introduction was completely rewritten, and more references were added to highlight ethnobotanical points relevant to the study area. See the new version of the paper.
Reviewer 3: Materials and Methods: The experimental design and the materials and methods section were Not adequately described. Please rewrite this section, use subheadings to organize the different methods, and describe all methods in a more detailed manner. Answer: The materials & methods were rewritten using subheadings to organize the different methods and to describe every section in a more detailed manner. A map was added to delimit our area of study and to show the languages/cultures distribution
Reviewer 3: Results: The authors did a good job in this section. The results were well presented But I think the author mixed up the discussion section here in the results although he made a separate section for the discussion that was poorly written. Answer: Results and discussion sections were merged. See the new version of the paper.
Reviewer 3: Discussion: I had a hard time making the connection between your discussions and the results in which figure. The author should mention data in figures and tables during the discussion. Please make subheadings in this section in an organized way, the same order as the subheadings in the results section. Please rewrite this section in a more organized way and relate to the data in tables and figures carefully with a comparison to the previous studies.*If it is easier to discuss the results clearly and thoroughly, the author is advised to combine the results and the discussion in one section for the best presentation and discussion to the results. Answer: Results and discussion sections were merged. See the new version of the paper. Thank you for your suggestions.
Reviewer 3: Conclusion: This section is missing. Although this section is not mandatory, it is better to add a conclusion section to this manuscript. Please provide a good conclusion for the study and include the significant findings. This section must be supported by the results. Answer: Conclusions were added. See new version.
Reviewer 3: References: Although the author provided enough citations, the author is advised to include more recent citations (The last 5 years' work). Answer: More references were added to highlight ethnobotanical points relevant to the study area. See the new version of the paper.
